behaviour/ecology

invasive species, native predator, squirrels, pine marten, scent cues, anti-predator behaviour

**Author for correspondence:**
Joshua P. Twining
e-mail: joshuaptwining@gmail.com

# Native and invasive squirrels show different behavioural responses to scent of a shared native predator

Joshua P. Twining[1], W. Ian Montgomery[1], Lily Price[1], Hansjoerg P. Kunc[1] and David G. Tosh[2]

[1]School of Biological Sciences, Queen's University of Belfast, 19 Chlorine Gardens, BT9 5DL Northern Ireland, UK
[2]National Museums NI, 153 Bangor Road, Cultra, BT18 0EU Northern Ireland, UK

JPT, 0000-0002-0881-9665; WIM, 0000-0001-9715-4767;
HPK, 0000-0003-4709-1352; DGT, 0000-0001-5210-6358

Invasive species pose a serious threat to native species. In Europe, invasive grey squirrels (*Sciurus carolinensis*) have replaced native red squirrels (*Sciurus vulgaris*) in locations across Britain, Ireland and Italy. The European pine marten (*Martes martes*) can reverse the replacement of red squirrels by grey squirrels, but the underlying mechanism of how pine martens suppress grey squirrels is little understood. Research suggests the reversal process is driven by direct predation, but why the native red squirrel may be less susceptible than the invasive grey squirrel to predation by a commonly shared native predator, is unknown. A behavioural difference may exist with the native sciurid being more effective at avoiding predation by the pine marten with which they have a shared evolutionary history. In mammals, olfactory cues are used by prey species to avoid predators. To test whether anti-predator responses differ between the native red squirrel and the invasive grey squirrel, we exposed both species to scent cues of a shared native predator and quantified the responses of the two squirrel species. Red squirrels responded to pine marten scent by avoiding the feeder, increasing their vigilance and decreasing their feeding activity. By contrast, grey squirrels did not show any anti-predator behaviours in response to the scent of pine marten. Thus, differences in behavioural responses to a shared native predator may assist in explaining differing outcomes of species interactions between native and invasive prey species depending on the presence, abundance and exposure to native predators.

# 1. Introduction

Invasive species are one of the greatest threats to biodiversity [1]. Invasive species have detrimental effects on native species arising from direct competition [2], predation [3] and/or introduction of disease [4]. The most severe impacts often occur following the introduction of novel invasive predators where naive native prey lack appropriate anti-predator responses to novel predators [3–7]. However, the reverse of this situation, where invasive prey species have established themselves in the absence of predators is less well documented. Such invasive species may lack appropriate anti-predator responses to subsequently recovering, native predator populations. This effect highlights the potential of recovering predator populations to provide biological control of predator naive, invasive species [8,9].

Prey species evolve predator avoidance mechanisms to reduce the risk of predation [7]. One method prey species use is to detect and respond to olfactory cues left by predators in their faeces or urine, which are typically used to mark territories and communicate with conspecifics [10,11]. These olfactory cues can reveal information about the presence of predators, allowing prey to exhibit anti-predator behaviours which minimize detection and/or direct contact, and thus reduce chances of predation [12,13]. The typical behavioural response of prey species to known predator odours is to alter space-use, movement, feeding and vigilance behaviour [14,15]. Research into how potential prey may respond to predation risk has a long history of using odour experiments. A widely practised method is the use of olfactory cues, in the form of urine or faeces, to simulate the presence of a predator [16–19]. Olfactory cues left by predators have distinct properties that make them especially pertinent during the search and detection phase of predator–prey interactions. Scent marks represent a spatially fixed olfactory signal, which can be especially useful to prey species as they provide information on locations of predictable predator activity [13]. Further, in contrast with other cues (i.e. visual or auditory) which are instantaneous, olfactory cues from scent marks can remain in the environment for a substantially longer duration, days [20,21] or even weeks [22]. Thus, being able to detect such cues and implement anti-predator behaviours can increase survival and fitness of prey. However, any direct increase in individual fitness due to anti-predator behaviours may be offset by significant foraging and reproductive costs [23]. Thus, prey species' response is not always predictable and varies based on environmental and temporal variables, as well as their history of encounters with the particular predator species [24–26].

Anti-predator behaviours typically evolve through a common evolutionary history between predator and prey, whereby sustained predation creates a sufficient selection pressure to drive the adaptation of such traits [27]. Therefore, a logical assumption is that invasive species occupying novel habitats, or native prey where an invasive predator with which they do not share such a common evolutionary history has appeared, may lack anti-predator behaviours [8,9,28–30]. However, the absence of anti-predator behaviours is not always as simple as the inability to recognize olfactory cues and failure to alter behaviour [26]. The response of naive prey species to novel predator odours can be nuanced, unpredictable and inconsistent, with some species not altering behaviour, while others have maladaptive responses such as being attracted to the predator's olfactory signals [28,31–36].

The eastern grey squirrel (*Sciurus carolinensis*), a North American native, has replaced the red squirrel (*Sciurus vulgaris*) across much of its former range in Great Britain and Ireland [37–39]. However, in certain parts of Ireland and Scotland, the presence of the European pine marten (*Martes martes*) can reverse the replacement of native red squirrels by invasive grey squirrels [40,41]. The underlying mechanism of this process is not well understood. Recent research suggests that it is likely to be driven by direct predation, although there is a paucity of data on the diet of pine martens where they co-occur with grey squirrels. This gap in our knowledge is likely to be due to the ephemeral nature of co-existence in the squirrel species [42]. Examples do exist, however, and demonstrate the grey squirrel to be a more frequently occurring prey item than the red squirrel when either of the species co-occur with the pine marten [43,44]. However, an explanation as to why two sciurid species that have similar ecologies would occur at different frequencies in the diet of a generalist opportunistic predator is yet to be elucidated. One potential explanation is that the two sciurid species respond differently to the presence of this commonly shared predator [41]. Both red and grey squirrels are preyed upon by the pine marten where they co-occur with the predator [43,45–49], but as a consequence of its co-evolution with the pine marten, the native red squirrel may have evolved anti-predator behaviours, while the invasive grey squirrels may lack such anti-predator behaviours, thus making them more vulnerable to predation.

To test whether free-living, wild red squirrels and grey squirrels differ in their anti-predator behaviour towards the pine marten, we exposed both squirrel species to olfactory cues from this commonly shared predator. By applying pine marten scent to squirrel feeding stations, we simulated the presence of pine marten and quantified behavioural differences between native and invasive

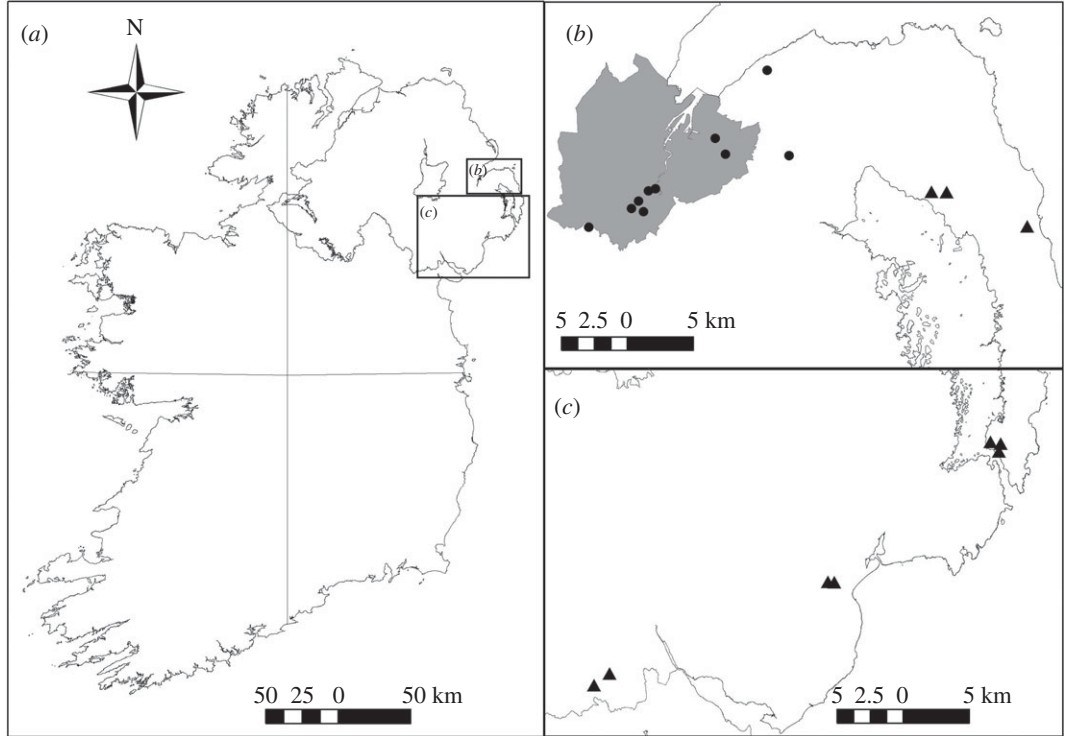

**Figure 1.** (*a*) A map showing location of Northern Ireland within Ireland; (*b*) map showing thirteen study sites in the Lagan valley and north Ards Peninsula in Co. Antrim and Co. Down; (*c*) map showing five study sites in south Ards Peninsula and the Mournes in Co. Down and two in the Ring of Gullion, Co. Armagh. Circles represent grey squirrel sites (*n* = 10), triangles represent red squirrel sites (*n* = 10).

squirrel species. We predicted that the invasive grey squirrel lacks anti-predator behavioural responses and would either not respond to pine marten scent (e.g. no change in number of visits or behaviours displayed) or demonstrate maladaptive responses (e.g. increasing visitation and feeding behaviour, decreasing vigilance displayed). By contrast, we predict that the native red squirrel alters its behaviour in response to the pine marten displaying anti-predator behaviours such as avoidance (e.g. reducing number of visits to the feeder where scent was applied), as well as decreasing feeding behaviour and increasing vigilance.

## 2. Material and methods

Data were collected from January to September 2015 at 20 sites in Northern Ireland (electronic supplementary material, table S1): allopatric populations of red (*n* = 10) and grey squirrels (*n* = 10) were sampled (figure 1). Habitats included coniferous plantations, deciduous forests, mixed coniferous and deciduous forests and suburban gardens (electronic supplementary material, Table S1). Red squirrels were largely sampled in coniferous plantations and mixed forests, and grey squirrels in mixed forests and suburban gardens due to species-specific habitat preferences and heterogeneous occurrence of either species within Northern Ireland. To compare behavioural responses between grey and red squirrels, a wooden squirrel feeder and a Bushnell HD Trophy Camera Trap were deployed at a single pre-selected, random location in each of the 20 sites. The squirrel feeders were custom made from softwood with a hinged, Perspex front behind which the bait sat (specifications and design: https://www.rsne.org.uk/sites/default/files/2018-05/Feeder%20Box%20Template.pdf). The bait was fully concealed by the feeder and the lid had to be lifted by the squirrel in order to access the bait, keeping bait dry and uncontaminated. Cameras were set at head height between 1 and 5 m from the feeder and were programmed to record 1 min videos when triggered by movement, with a 1 s reset time. Feeders were filled with a 50:50 mix of peanuts and sunflower seeds, and bait was replenished weekly. The experiment was conducted over two weeks at each site, with pine marten scent applied to each feeder once after the first week. Pine marten scent was derived from mixing a fresh pine marten scat which had been stored at −20°C from the day of collection and then defrosted prior to application. Fresh pine marten scats were identified using established methodology [50].

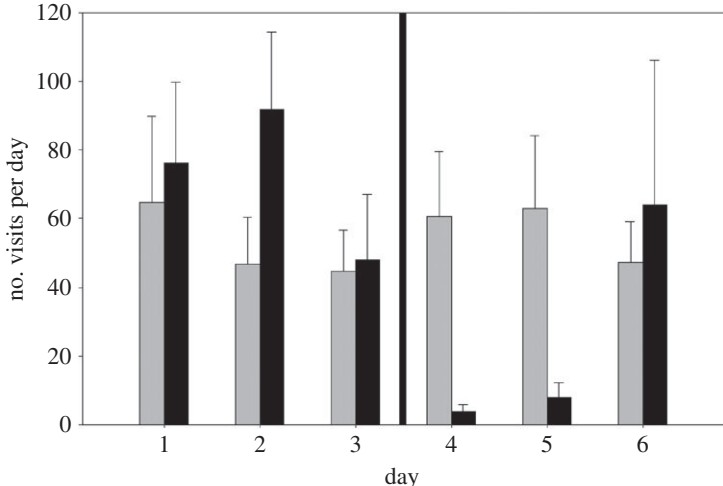

**Figure 2.** Mean (95% clm) number of daily visits to feeders by red squirrels (black bars) and grey squirrels (grey bars) three days prior, and three post application of pine marten scent. The dark vertical line illustrates when the pine marten scent was applied.

A pine marten scat was mixed in 500 ml of water and this solution applied to the external wooden parts of the feeder only using a spray bottle ensuring food was not contaminated.

To compare behavioural changes in response to pine marten scent, we analysed the behaviour of animals starting one week prior to application of scent, and one week immediately following the application of scent. Videos were viewed using the freeware VLC media player (https://www.videolan.org/vlc/). The following response variables with respect to squirrel behaviour were recorded: number of visits (per day) and duration of each visit to the feeder (s). For each visit, we calculated the proportion of time spent performing different behaviours. The two most important behaviours to this study were feeding (%) and the proportion of time being vigilant (%) [51]. Feeding and vigilance are well-established behavioural indices for providing non-invasive quantification of fear response [52,53]. Feeding was typically performed in a hunched posture, with front limbs being used to hold food up to a downwards facing head, or alternatively the squirrel would take a quadrupedal posture eating straight from the ground/feeder, and head downward facing. Mastication of food was very pronounced in either stance. Vigilance was recognized when individuals were in a bipedal posture with their head raised and rotating frequently, visually scanning their surroundings [52–55]. Tail flagging is also a recognized part of vigilance in response to a potential threat [51,54].

A total of 10 897 visits were recorded over 280 recording days totalling 8197 min of recordings (grey squirrels: 6076 visits totalling 3816 min; red squirrels: 4741 visits totalling 4363 min) at twenty sites. Two-day periods pre- and post-treatment were used for the statistical analysis as the application of pine marten scent was observed to have a 48 h efficacy on behavioural responses in the squirrel species (figure 2). Analysis of behaviour, therefore, was based on data collected over 80 days from 20 sites totalling 4178 min of recordings (greys squirrels: 2150 visits totalling 1218 min; red squirrels: 2142 visits totalling 2960 min). Red squirrels did not return to four of the sites after application of pine marten scent. These sites were not able to be used in the proportions of time spent feeding and vigilant aspect of the analysis. Additionally, image quality was not sufficient at two of the sites to distinguish behaviours consistently and these sites were also discarded from this part of the analysis.

Number of visits, duration of visits and proportions of time spent feeding and vigilant were modelled against two-day periods prior to and after treatment with pine marten scent, using generalized mixed effects models (GLMMs) with Poisson errors and site as a random effect. All response variables were modelled as a function of treatment, species and the interaction between the two. Analyses were performed using R v. 3.2.1 with nlme package [55,56].

## 3. Results

Number of visits per day declined after scent application in red squirrels but not in grey squirrels (figure 3a; GLMM: treatment: $F_{1,61} = 12.112$, $p \leq 0.001$; species: $F_{1,61} = 34.816$, $p = 0.773$; interaction between species and treatment: $F_{1,61} = 25.157$, $p \leq 0.001$). Similarly, the duration of visits decreased in red squirrels but not grey squirrels after scent application (figure 3b, GLMM: treatment: $F_{1,3379} =$

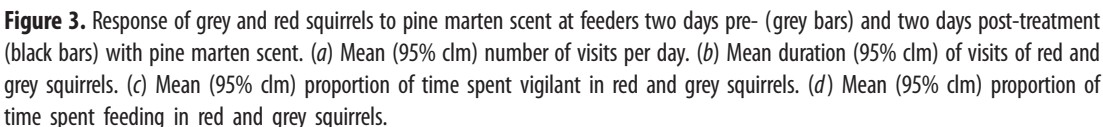

**Figure 3.** Response of grey and red squirrels to pine marten scent at feeders two days pre- (grey bars) and two days post-treatment (black bars) with pine marten scent. (*a*) Mean (95% clm) number of visits per day. (*b*) Mean duration (95% clm) of visits of red and grey squirrels. (*c*) Mean (95% clm) proportion of time spent vigilant in red and grey squirrels. (*d*) Mean (95% clm) proportion of time spent feeding in red and grey squirrels.

28.372, $p \leq 0.001$; species: $F_{1,18} = 1.637$, $p = 0.217$; interaction between species and treatment: $F_{1,3379} = 64.302$, $p \leq 0.001$). Moreover, red squirrels were more vigilant after scent application, while conversely grey squirrels were less vigilant (figure 3*c*, GLMM: treatment: $F_{1,15} = 5.166$, $p = 0.044$; species: $F_{1,15} = 2.73$; $p = 0.118$; interaction between species and treatment: $F_{1,15} = 23.76$, $p \leq 0.001$). Finally, red but not grey squirrels spent significantly less time feeding after scent application (figure 3*d*, GLMM: treatment: $F_{1,15} = 10.679$, $p = 0.005$; species: $F_{1,15} = 8.9711$, $p = 0.009$; interaction between species and treatment: $F_{1,15} = 11.704$, $p = 0.005$). See table 1 for breakdown of statistical results for other non-significant behaviours.

## 4. Discussion

The patterns of response to olfactory cues of a shared native predator differed between native red squirrels and invasive grey squirrels. After the application of pine marten scent, red squirrels reduced both visitation rates and the duration of visits to feeders, whereas grey squirrels did not significantly change their visitation rate. At some sites, red squirrels did not return to the feeder for a 48 h period. This provides further evidence of avoidance to olfactory cues of a native predator in the native red squirrel, but not in the invasive grey squirrels.

A common anti-predator response in prey species is to increase vigilance when predators or signs of predators are present [57]. We found that red squirrels were more vigilant and decreased feeding after the application of pine marten scent while grey squirrels conversely decreased vigilance, while not significantly altering feeding behaviour. This difference between the two species suggests that red squirrels detect the olfactory cues of the pine marten and perceive them as a risk. This increase in vigilance corresponds with a decrease in feeding. Thus, while red squirrels decrease the risk of being predated, they are less effective in exploiting a food source than the grey squirrels. Grey

**Table 1.** The effect of the presence of pine marten scent on the behavioural responses of grey and red squirrels. Results are obtained from generalized mixed effect models. Italics indicate significant results.

| species | behaviour | d.f. | $F_2$ | p |
|---|---|---|---|---|
| red squirrel | feeding | 4 | 8.417 | *0.044* |
| | vigilance | 4 | 14.729 | *0.019* |
| | movement | 4 | 0.394 | 0.564 |
| | investigatory | 4 | 0.643 | 0.467 |
| | social | 4 | 1.923 | 0.237 |
| | aggression | 4 | 0.263 | 0.635 |
| grey squirrel | feeding | 7 | 2.193 | 0.182 |
| | vigilance | 7 | 6.786 | *0.035* |
| | movement | 7 | 0.260 | 0.625 |
| | investigatory | 7 | 0.0002 | 0.988 |
| | social | 7 | 1.00 | 0.351 |
| | aggression | 7 | 0.056 | 0.819 |

squirrels, on the other hand, do not seem to recognize the scent of the pine marten as a risk and actually decreased vigilance after application of pine marten scent. This is in line with other studies which have reported that naive prey species rather than avoiding predator scents can be drawn to them [33,58].

The difference in response to pine marten scent between the two squirrel species suggests that behavioural differences may allow the native species to avoid native predators more successfully. This is in line with findings in recolonizing predators and naive prey: brown bears (*Ursus arctos*) recolonizing Scandinavia exhibited disproportionally high predation rates on naive elk [59]. This study on elk, although on species that have previously co-occurred, suggests a lack of anti-predator behaviours can result in higher levels of predation until anti-predator behaviours are readopted. Thus, evolutionarily naive grey squirrels may be more vulnerable to predation by pine martens than red squirrels due to a lack of anti-predator behaviours in response to pine marten olfactory cues. Red squirrels typically occur in the diet of pine martens at low frequencies, whereas grey squirrels occur at significantly higher frequencies [45–49]. Thus, a driver of the reversal of red squirrel replacement by grey squirrels [40,41] could be direct predation [43,44], which is in part explained by the differences in the response to scent cues of a shared native predator in the invasive grey squirrels compared to the native red squirrels.

The lack of anti-predator behavioural response in the invasive grey squirrel to the native predator, the pine marten, may only be temporary. While prey species typically develop anti-predator responses through coevolution with predators, prey facing strong selection pressures can develop anti-predator behaviours within a single generation e.g. moose (*Alces alces*) developing hyper-vigilance in response to wolves (*Canis lupus*) [60], or within a few generations e.g. ring-tailed possums (*Pseudocheirus peregrinus*) responding to olfactory cues of alien red foxes (*Vulpes vulpes*) [19]. However, the development of effective anti-predator responses is not guaranteed, and many naive species have become extinct due to introduction of novel predators prior to the development of effective anti-predator responses [6,19]. Thus, the long-term lack of behavioural response observed in grey squirrels to pine marten remains equivocal.

Olfactory cues remain detectable for a prolonged period of time in contrast with other sensory modalities, e.g. visual/acoustic cues. The behavioural response of red squirrels to pine marten scent degenerated over 48–72 h (figure 2). These results provide further evidence of the ability of prey species to discriminate the age of an olfactory signal of predators and use this information in determining trade-offs between high value foraging and predation risk, with behaviours returning to pre-scent application levels after 48 h [26]. The deterioration effect may be stronger in our experimental exposure than in a natural setting, as the scats were dissolved in water, which may have altered the chemical compounds. However, the present results are in line with previous work on prey species' response to chemo-olfactory cues of larger predators [21,61].

Scent of only one predator species was used in this experiment. The pine marten is the sole arboreal predator in Ireland and thus the most likely to be encountered by red squirrels. However, it cannot be confirmed unequivocally that the red squirrels were specifically responding to the scent of the pine marten, a native predator, and not just responding to any scent. Due to the fact that the invasive prey species failed to demonstrate anti-predator behaviours in response to the olfactory signal of the native predator, in contrast with the native prey species, familiarity specifically with pine marten scent was probably important in determining response of the latter [26]. To confirm these results, future research should be conducted with multiple scent cues including: a native predator, a conspecific or non-predatory scent (i.e. Rabbit, *Oryctolagus cuniculus*) and a control. This would provide further information on the prey species' sensitivity to differing scents and allow further and more accurate dissection of specificity of behavioural responses to differing olfactory signals.

In conclusion, we demonstrate different patterns of response to olfactory cues of a shared predator in an invasive and a native squirrel species. The marked changes in behavioural response of the native red squirrel after the presentation of olfactory cues to a native predator validate our experimental approach. The lack of anti-predator behaviours observed in the invasive grey squirrel provides a potential explanation for higher predation rates of grey squirrels than red in the diet of the pine marten [49,50]. Our results suggest that the observed ability of recovering native predator populations to provide biological resistance to invasive species is in part due to their failure to recognize the olfactory signals of these predators.

Ethics. This study was assessed as being below the threshold of the Animals (Scientific Procedure) Act 1986 (ASPA) as it was non-invasive involving only behavioural monitoring. Due to regulations put in place after this study was conducted requiring ethical approval for non-ASPA-related research, the chair of Queen's University, Belfast School of Biological Sciences Animal Research Ethics committee retrospectively assessed the study and methods, confirming there are no ethical issues with the study.

Data accessibility. All data for the project and all the R code used in analysis and production of figures can be found here: https://doi.org/10.5061/dryad.8cz8w9gkf.

Authors' contributions. J.P.T. conducted all behavioural analysis, statistical analysis, wrote the first draft of the manuscript and worked on revisions of the manuscript. W.I.M. assisted with analysis and both first draft and revisions of the manuscript. L.P. conducted the fieldwork. D.G.T. conceptualized the project, attained funding, conducted fieldwork and assisted with production and revisions of manuscript; H.P.K. conceptualized the project, assisted with behavioural and statistical analysis and assisted with production and revisions of manuscript.

Competing interests. We have no competing interests.

Funding. This projects funding was entirely sourced through crowd-funding on www.kickstarter.com, see https://www.kickstarter.com/projects/318099790/the-squirrel-predator-prey-project for project details. The lead author is a PhD student at Queen's University, Belfast, their studentship is funded by Department of Learning in Northern Ireland.

Acknowledgments. We would like to thank those that contributed to the Kickstarter campaign that funded the behavioural project, in particular the Vincent Wildlife Trust Ireland and Belos Ecology. In addition, we are grateful to landowners who allowed this work to be conducted on their property, the Department of Learning in Northern Ireland for funding the studentship; Aoden Matthews and Joshua Clarke for work as research assistants. We would like to thank our two anonymous reviewers and Dr Emma Sheehy, who provided useful feedback and constructive criticisms allowing us to improve our manuscript substantially. Finally, thank you to Jennifer Anson, whose work in Australia provided the inspiration for this study.

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
