## [Reviewer comments · Royal Society Open Science]

Review History

RSOS-191841.R0 (Original submission)

Review form: Reviewer 1 (Emma Sheehy)

Is the manuscript scientifically sound in its present form?

Yes

Are the interpretations and conclusions justified by the results?

Yes

Is the language acceptable?

Yes

Do you have any ethical concerns with this paper?

No

Have you any concerns about statistical analyses in this paper?

No

Recommendation?

Accept as is

Comments to the Author(s)

I am satisfied that the authors have addressed both my own and other reviewers queries and comments and am pleased to recommend that the manuscript be accepted for publication. Emma

Review form: Reviewer 2

Is the manuscript scientifically sound in its present form?

No

Are the interpretations and conclusions justified by the results?

No

Is the language acceptable?

Yes

Do you have any ethical concerns with this paper?

Yes

Have you any concerns about statistical analyses in this paper?

Yes

Recommendation?

Reject

Comments to the Author(s)

What was a relatively small matter in relation to ethics, since in my opinion the work is non-invasive and consent would readily have been granted by most ethics committees, has become more problematic since it appears the work, although non-invasive, did not comply with local regulations, which require appraisal by the school ethics committee. The time required to obtain such approval is no justification for not having it.

I previously raised issues with experimental design. Principally that there was a lack of a with and without comparison, only before and after, and that since only extract of marten scent was applied, it cannot be concluded that it was not scent of any kind that elicited the observed response. I should say that I am inclined to agree with the authors in respect of their interpretation of the results. However, the formalities of such behavioural experimentation are that they cannot exclude other possibilities with this design, i.e. that conditions on the days after application of scent differed and reduced visitations, or that any unpleasant or novel scent, or any scent at all (not necessarily that of a predator or of martens) would have elicited the same response.

They do now acknowledge the limitation of using only one scent, and this is supported by a narrative recommending future studies. I would point out that my critique was that the treatment consisted of application of marten scent, but it could be that the response by the squirrels could, in principle have been to any scent. Thus the proposed designs do not wholly address the problem.

In the authors' response to my comment about before-after but not with-without comparisons, they include a summary of variation between days and cite a series of F-statistics and refer to a Figure 5. I cannot find a Figure 5 in the main manuscript or supplementary materials. Do they mean Figure 2? They highlight that days 1-3, 6 and 7, do not differ from one another, but that 4

and 5 do differ from the others. At this point they state that day 4 differs from the others with $p=0.083$. Does this mean they increased alpha (perhaps in post-hoc testing?) for significance here to 0.1? This would need some justification. Or is it a typo, as the mean for this day appears lower than that of the day after?

Decision letter (RSOS-191841.R0)

21-Jan-2020

Dear Mr Twining,

I am writing in regards to your manuscript submitting to Royal Society Open Science.

On behalf of the Editors, I am pleased to inform you that your Manuscript RSOS-191841 entitled "Native and invasive squirrels show different behavioural responses to scent of a shared native predator" has been accepted for publication in Royal Society Open Science subject to minor revision in accordance with the referee suggestions. Please find the referees' comments at the end of this email.

The reviewers and handling editors have recommended publication, but also suggest some minor revisions to your manuscript. Therefore, I invite you to respond to the comments and revise your manuscript. Additionally, we ask that you please also upload the following to your revised submission:

- Your revised manuscript text which clarifies that the scent was applied only to the external surface of the feeders - from the Editors' reading of the text, this is still only implied rather than stated.
- Please provide the documentation obtained regarding the ethical approval as a "Response to Referees" letter.
- A point-by-point response to the reviewers, as well as the Associate Editors' comments below; uploaded as another "Response to Referees" letter.

Because the schedule for publication is very tight, it is a condition of publication that you submit the revised version of your manuscript before 30-Jan-2020. Please note that the revision deadline will expire at 00.00am on this date. If you do not think you will be able to meet this date please let me know immediately.

In order to expedite the processing of the revised manuscript, please be as specific as possible in your response to the referees. We strongly recommend uploading two versions of your revised manuscript:

- 1) Identifying all the changes that have been made (for instance, in coloured highlight, in bold text, or tracked changes);

If your manuscript is newly submitted and subsequently accepted for publication, you will be asked to pay the article processing charge, unless you request a waiver and this is approved by Royal Society Publishing. You can find out more about the charges at <https://royalsocietypublishing.org/rsos/charges>. Should you have any queries, please contact openscience@royalsociety.org.

Kind regards,
Lianne Parkhouse
Editorial Coordinator
Royal Society Open Science
openscience@royalsociety.org

on behalf of Dr Punidan Jeyasingh (Associate Editor) and Professor Kevin Padian (Subject Editor)
openscience@royalsociety.org

Associate Editor Comments to Author (Dr Punidan Jeyasingh):

I thank the authors for the patience and prompt support in helping us understand the situation. The documentation provided and the minor tweaks to the manuscript makes it clear that the study was in compliance with animal care regulations. I am happy to recommend it for publication.

I wish to share this confidential note from a previous reviewer. I felt it was a fair point, and hope the authors will consider it (hence the accept with minor revision recommendation).

"The authors have acknowledged two male professors thanking them for providing comments on earlier versions of the manuscript. Given the manuscript has been through multiple journals, iterations, and is now at a version that is far removed from any earlier versions of the manuscript, I would find it more appropriate for the authors to acknowledge the work of the reviewers who have spent comparatively much more time and effort on bringing this manuscript to publishable standard than two professors who realistically, given the standard of the earliest version, could not have spent more than a few minutes glancing over it!"

Reviewer comments to Author:

Reviewer: 1

Comments to the Author(s)

I am satisfied that the authors have addressed both my own and other reviewers queries and comments and am pleased to recommend that the manuscript be accepted for publication. Emma

Reviewer: 2

Comments to the Author(s)

What was a relatively small matter in relation to ethics, since in my opinion the work is non-invasive and consent would readily have been granted by most ethics committees, has become more problematic since it appears the work, although non-invasive, did not comply with local regulations, which require appraisal by the school ethics committee. The time required to obtain such approval is no justification for not having it.

I previously raised issues with experimental design. Principally that there was a lack of a with and without comparison, only before and after, and that since only extract of marten scent was applied, it cannot be concluded that it was not scent of any kind that elicited the observed response. I should say that I am inclined to agree with the authors in respect of their interpretation of the results. However, the formalities of such behavioural experimentation are that they cannot exclude other possibilities with this design, i.e. that conditions on the days after application of scent differed and reduced visitations, or that any unpleasant or novel scent, or any scent at all (not necessarily that of a predator or of martens) would have elicited the same response.

They do now acknowledge the limitation of using only one scent, and this is supported by a narrative recommending future studies. I would point out that my critique was that the treatment consisted of application of marten scent, but it could be that the response by the squirrels could, in principle have been to any scent. Thus the proposed designs do not wholly address the problem.

In the authors' response to my comment about before-after but not with-without comparisons, they include a summary of variation between days and cite a series of F-statistics and refer to a Figure 5. I cannot find a Figure 5 in the main manuscript or supplementary materials. Do they

mean Figure 2? They highlight that days 1-3, 6 and 7, do not differ from one another, but that 4 and 5 do differ from the others. At this point they state that day 4 differs from the others with $p=0.083$. Does this mean they increased alpha (perhaps in post-hoc testing?) for significance here to 0.1? This would need some justification. Or is it a typo, as the mean for this day appears lower than that of the day after?

Author's Response to Decision Letter for (RSOS-191841.R0)

See Appendices A & B.

Decision letter (RSOS-191841.R1)

27-Jan-2020

Dear Mr Twining,

It is a pleasure to accept your manuscript entitled "Native and invasive squirrels show different behavioural responses to scent of a shared native predator" in its current form for publication in Royal Society Open Science. The comments of the reviewer(s) who reviewed your manuscript are included at the foot of this letter.

Please note: we require every author to have an active email address. At present 'lprice01@qub.acuk' is not accepting messages from Royal Society Open Science - please can you ensure an active email address is sent to the Editorial Office urgently?

on behalf of Dr Punidan Jeyasingh (Associate Editor) and Kevin Padian (Subject Editor)
openscience@royalsociety.org

Associate Editor Comments to Author (Dr Punidan Jeyasingh):

Associate Editor

Comments to the Author:

I am happy to recommend this manuscript for publication. I thank the authors for their efficient job incorporating comments and concerns throughout, and for their patience as we sorted out the animal ethics issue.

Appendix A

Response to reviewers

We would like to thank all three reviewers, the associate editor and the subject editor for their time and effort in reviewing our manuscript and allowing us to improve it to the point of acceptance for publication at Royal Society Open Science.

Associate Editor Comments to Author (Dr Punidan Jeyasingh):

I thank the authors for the patience and prompt support in helping us understand the situation. The documentation provided and the minor tweaks to the manuscript makes it clear that the study was in compliance with animal care regulations. I am happy to recommend it for publication.

I wish to share this confidential note from a previous reviewer. I felt it was a fair point, and hope the authors will consider it (hence the accept with minor revision recommendation).

"The authors have acknowledged two male professors thanking them for providing comments on earlier versions of the manuscript. Given the manuscript has been through multiple journals, iterations, and is now at a version that is far removed from any earlier versions of the manuscript, I would find it more appropriate for the authors to acknowledge the work of the reviewers who have spent comparatively much more time and effort on bringing this manuscript to publishable standard than two professors who realistically, given the standard of the earliest version, could not have spent more than a few minutes glancing over it!"

We thank the editor for providing us with this comment. We completely agree with the reviewer, as such we have replaced previous thanks to professors *with* “We would like

to thank our two anonymous reviewers and Dr Emma Sheehy, who provided useful feedback and constructive criticisms allowing us to improve our manuscript substantially.”

Reviewer comments to Author:

Reviewer: 1

Comments to the Author(s)

I am satisfied that the authors have addressed both my own and other reviewers queries and comments and am pleased to recommend that the manuscript be accepted for publication.

Emma

Thank you Emma, we really appreciate all the time and effort you put into reviewing this manuscript in its different forms and helping to greatly improve the manuscript prior to publication.

Reviewer: 2

Comments to the Author(s)

What was a relatively small matter in relation to ethics, since in my opinion the work is non-invasive and consent would readily have been granted by most ethics committees, has become more problematic since it appears the work, although non-invasive, did not comply with local regulations, which require appraisal by the school ethics committee. The time required to obtain such approval is no justification for not having it.

Local regulations cited were not in place when fieldwork was conducted in 2015, we went through all the correct channels. This has now been clarified and confirmed with the editors to their satisfaction.

I previously raised issues with experimental design. Principally that there was a lack of a with and without comparison, only before and after, and that since only extract of marten scent was applied, it cannot be concluded that it was not scent of any kind that elicited the observed response. I should say that I am inclined to agree with the authors in respect of their interpretation of the results. However, the formalities of such behavioural experimentation are that they cannot exclude other possibilities with this design, i.e. that conditions on the days after application of scent differed and reduced visitations, or that any unpleasant or novel scent, or any scent at all (not necessarily that of a predator or of martens) would have elicited the same response.

They do now acknowledge the limitation of using only one scent, and this is supported by a narrative recommending future studies. I would point out that my critique was that the treatment consisted of application of marten scent, but it could be that the response by the squirrels could, in principle have been to any scent. Thus the proposed designs do not wholly address the problem.

We appreciate the reviewer's clarification and have now revised our suggestion further. To address the issue of scent sensitivity and concerns of control (although we still believe each day previous to addition of pine marten scent acted as a control in this experiment), we have amended our future research suggestion to the following:

“Scent of only one predator species was used in this experiment. The pine marten is the sole arboreal predator in Ireland and thus the most likely to be encountered by red squirrels. However, it cannot be confirmed unequivocally that the red squirrels were

specifically responding to the scent of the pine marten, a native predator, and not just responding to any scent. Due to the fact that the invasive prey species failed to demonstrate anti-predator behaviours in response to the olfactory signal of the native predator, in contrast to the native prey species, familiarity specifically with pine marten scent was likely important in determining response of the latter [26]. To confirm these results future research should be conducted with multiple scent cues including: a native predator, a conspecific or non-predatory scent (i.e. Rabbit, *Oryctolagus cuniculus*) and a control. This would provide further information on the prey species' sensitivity to differing scents and allow further and more accurate dissection of specificity of behavioural responses to differing olfactory signals. "

In the authors' response to my comment about before-after but not with-without comparisons, they include a summary of variation between days and cite a series of F-statistics and refer to a Figure 5. I cannot find a Figure 5 in the main manuscript or supplementary materials. Do they mean Figure 2? They highlight that days 1-3, 6 and 7, do not differ from one another, but that 4 and 5 do differ from the others. At this point they state that day 4 differs from the others with $p=0.083$. Does this mean they increased alpha (perhaps in post-hoc testing?) for significance here to 0.1? This would need some justification. Or is it a typo, as the mean for this day appears lower than that of the day after?

We thank the reviewer again for their thorough reading of the manuscript and our responses. We do apologise, the response to the question was copied from an earlier version of the manuscript (hence referring to Figure 5, when in this manuscript it is Figure 2 as the reviewer deduced), additionally the reviewer is correct in their supposition, the P value result for Day 4 was a typo missing a zero and the results for day four at three decimal places should have been reported as $P = 0.008$).

Appendix B

Dr Gareth Arnott
Senior Lecturer Animal Behaviour and
Welfare
Queen's University Belfast
School of Biological Sciences
Medical Biology Centre
97 Lisburn Road
Belfast BT9 7BL
UK
Email: g.arnott@qub.ac.uk

19/12/2019

To whom it may concern,

I am writing in my role as Chair of a university Faculty level sub-committee for animal research ethics, and as former Chair of the School of Biological Sciences Animal Research Ethics committee at Queen's University Belfast.

This is in response to a paper submitted to your journal by Mr Twining entitled "Native and invasive squirrels show different behavioural responses to scent of a shared native predator".

I understand that there is a query regarding the ethical approval for this application. I have discussed this with the authors and at the time of data collection the project did not require ethical approval under local regulations. This was because their study had previously been assessed as being below the threshold of the Animals (Scientific Procedures) Act 1986 (ASPAs). Regulations requiring ethical approval at the school level for non-ASPAs related animal research were introduced in 2016, a year after the data for this study were collected.

To deal with this, in my role I have now retrospectively assessed their study and methods. Having done this, I can confirm that there are no ethical issues with the study as it was non-invasive, involving behavioural monitoring. It would have passed our assessments at the school committee level. Therefore, I see no reason why it should not be published in your journal.

Should you require any additional information then please do not hesitate to contact me.

Yours faithfully,

Dr Gareth Arnott